# Global Terrapin Character-Based DNA Barcodes: Assessment of the Mitochondrial COI Gene and Conservation Status Revealed a Putative Cryptic Species

**DOI:** 10.3390/ani13111720

**Published:** 2023-05-23

**Authors:** Mohd Hairul Mohd Salleh, Yuzine Esa, Rozihan Mohamed

**Affiliations:** 1Department of Aquaculture, Faculty of Agriculture, Universiti Putra Malaysia, Serdang 43400, Malaysia; 2Royal Malaysian Customs Department, Persiaran Perdana, Presint 2, Putrajaya 62596, Malaysia; 3International Institute of Aquaculture and Aquatic Sciences, Universiti Putra Malaysia, Lot 960 Jalan Kemang 6, Port Dickson 71050, Malaysia

**Keywords:** Southern River Terrapin, genetics, haplotype, phylogenetic tree, Peninsular Malaysia, population diversity

## Abstract

**Simple Summary:**

This study evaluated 26 sequences of terrapins worldwide through COI DNA barcoding and phylogenetic analysis, which included 12 species and three families. Moreover, 16 haplotypes were found; they were either misidentified, or a potential cryptic species was determined between *B. baska* and *B. affinis affinis*. Thus, COI remains an effective barcode marker for the terrapin species.

**Abstract:**

Technological and analytical advances to study evolutionary biology, ecology, and conservation of the Southern River Terrapin (*Batagur affinis* ssp.) are realised through molecular approaches, including DNA barcoding. We evaluated the use of COI DNA barcodes in Malaysia’s Southern River Terrapin population to better understand the species’ genetic divergence and other genetic characteristics. We evaluated 26 sequences, including four from field specimens of Southern River Terrapins obtained in Bota Kanan, Perak, Malaysia, and Kuala Berang, Terengganu, Malaysia, as well as 22 sequences from global terrapins previously included in the Barcode of Life Database (BOLD) Systems and GenBank. The species are divided into three families: eight Geoemydidae species (18%), three Emydidae species (6%), and one Pelomedusidae species (2%). The IUCN Red List assigned the 12 species of terrapins sampled for this study to the classifications of critically endangered (CR) for 25% of the samples and endangered (EN) for 8% of the samples. With new haplotypes from the world’s terrapins, 16 haplotypes were found. The intraspecific distance values between the COI gene sequences were calculated using the K2P model, which indicated a potential cryptic species between the Northern River Terrapin (*Batagur baska*) and Southern River Terrapin (*Batagur affinis affinis*). The Bayesian analysis of the phylogenetic tree also showed both species in the same lineage. The BLASTn search resulted in 100% of the same species of *B. affinis* as *B. baska*. The Jalview alignment visualised almost identical sequences between both species. The Southern River Terrapin (*B. affinis affinis*) from the west coast of Peninsular Malaysia was found to share the same haplotype (Hap_1) as the Northern River Terrapin from India. However, *B. affinis edwardmolli* from the east coast of Peninsular Malaysia formed Hap_16. The COI analysis found new haplotypes and showed that DNA barcodes are an excellent way to measure the diversity of a population.

## 1. Introduction

Terrapins inhabit either freshwater or brackish water [1]. There is no clear taxonomic group for terrapins, which may be unrelated. Numerous species belong to the families of Geoemydidae and Emydidae [2]. The only terrapin species not in this group is the *Pelusios seychellensis* from Seychelles [3].

The “Barcode of Life” Consortium is a global effort to conduct a molecular inventory of the planet’s biodiversity [4]. After it was demonstrated that the cytochrome c oxidase subunit I (COI) gene of the mitochondrial DNA (mtDNA) could be used to successfully identify North American bird species, such as *Sturnella magna*, *Tringa solitaria*, and *Hirundo rustica* [5], numerous other vertebrate COI barcodes have been developed [6,7,8]. Ref. [9] also reported that the COI marker was better for barcoding than sequences from the mitochondrial control region.

Traditional taxonomy frequently fails to distinguish between the different terrapin species because they lack essential morphological characteristics. Currently, molecular methods are required to identify certain species [10,11]. A complementing tool to traditional taxonomy and systematics research, DNA barcoding allows for a more accurate understanding of the existing fauna around the world [12]. Especially in species with complicated, accessible anatomy, DNA barcoding is proposed as a method for quickly and readily identifying species using a short DNA sequence [12,13]. DNA barcoding has been used to identify freshwater turtles all over the world, even in Malaysia [14].

*Batagur affinis* ssp. [15] is among 24 species of turtles found in Peninsular Malaysia [16] and Sumatra, Indonesia, and was initially believed to be conspecific with *B. baska*, a species native to the North (Bangladesh and India) [17]. According to [18], *B. baska* consisted of at least two heritably distinct species: *B. affinis* ssp. populations in the Kedah River systems and *B. affinis affinis* populations in the Perak River systems, both on the west coast of Peninsular Malaysia. In contrast, individuals in the Terengganu River basin were identified as *B. affinis edwardmolli*. According to [19], this species is one of the world’s 25 most endangered freshwater turtles and tortoises.

*B. affinis* ssp. used to live in a large river in Southeastern Asia, including the Tonle Sap in Cambodia and the Mekong delta in Vietnam. However, many of its wild populations have been severely reduced or wiped out [20,21,22,23,24]. *Batagur affinis* ssp. is found only on the west coast of Peninsular Malaysia and is extinct in Sumatra, Indonesia [25,26].

In contrast, the subspecies *B. a. edwardmolli*, located on the east coast of Peninsular Malaysia that once reached from Singapore to Southeast Asia, is now thought to have vanished from Vietnam, Thailand, Singapore, and Indonesia [23,26,27]. Currently, only Peninsular Malaysia and Cambodia are home to this species [18,23,24,28]. Moreover, according to [23], there are still populations of *B. a. edwardmolli* in Cambodia and along the east coast of Peninsular Malaysia. This implies that the Malaysian and Cambodian populations are the only ones whose genes have remained constant across the species’ range.

Unfortunately, this study was carried out during a difficult period, namely the COVID-19 pandemic. Due to the Malaysian Movement Control Order (MCO), or lockdown, we were only permitted to gather four specimens of the Southern River Terrapin from Peninsular Malaysia by the Malaysian government authority. The samples are limited due to the conservation status of *B. affinis* ssp., which has been listed as critically endangered on the IUCN Red List since 2000 [16]. This study compares them to the other eleven terrapin species listed by [3,26] and accessed from the public database portal.

In addition, we were the first to upload COI *B. affinis* ssp. sequences to the GenBank database portal. The objectives of this study were to determine if terrapin DNA barcoding could be used all over the world by comparing the unique COI sequences to other COI sequences that were already available from the Barcode of Life Data (BOLD) Systems and GenBank, and to analyse the phylogenetic relationships among terrapins, including the recently collected specimens from Malaysia.

## 2. Materials and Methods

### 2.1. Study Sites

Four *Batagur affinis* ssp. individuals from two distinct population locations on the east and west coasts of Peninsular Malaysia were randomly chosen for this study, and the sampling was carried out in 2020 (Figure 1). The captive hatchling population at the Bota Kanan head-starting facility (BK; GPS coordinates: 4.3489° N and 100.8802° E) in Perak, Malaysia, provided the blood samples of *B. affinis affinis* (N = 1). The facilities were developed beside the Perak River, which is a habitat for the wild Southern River Terrapin population. There was no uncertainty regarding the genetic origin of that sample. In addition, blood samples from three wild *B. affinis edwardmolli* hatchlings (translocated eggs) were taken from a population in Bukit Paloh, Kuala Berang (KB; GPS coordinates: 5.0939° N, 102.7821° E), which is in Terengganu, Malaysia. According to [29], blood was drawn from the species using venipuncture methods through the internal jugular vein and subcarapacial venous plexus (SVP). In a 2 mL microcentrifuge tube, 1.5 mL of blood was preserved with 0.5 mL of EDTA in a 1:3 ratio before being kept at −20 °C. The Department of Wildlife and National Parks, Peninsular Malaysia, issued the study and field permit approval number, which is B-00335-16-20.

### 2.2. DNA Isolation, PCR, and Sequencing

For each sample, 200 µL of EDTA whole blood was used to extract the nucleic acids. After cell lysis and protein denaturation, DNA was extracted using the ReliaPrep^TM^ Blood gDNA Miniprep System with binding column technology (Promega, Madison, WI, USA) according to the manufacturer’s instructions. The final volume extracted was adjusted to 200 µL based on the input volume of the EDTA whole-blood sample. Using the Thermo Scientific^TM^ NanoDrop 2000 c spectrophotometer model ND-2000, the amount and purity of the extracted DNA samples were evaluated (Thermo Fisher Scientific, Waltham, MA, USA). After quantifying the extracted nucleic acids, the DNA samples were put onto a 1% (*w*/*v*) agarose gel with molecular markers. Electrophoresis was performed to assess the integrity and intactness of the high molecular weight DNA band.

The cross-species primer derived from Painted Terrapin, *Batagur borneoensis*, was utilised for PCR. Ref. [30] made the “Tuntong” primer pair, which targets the COI marker gene. The forward primer (5-CGCGGAATTAAGCCAACCAG-3) and the reverse primer (5-TTGGTACAGGATTGGGTCGC-3) are designed. The COI gene fragment PCR amplification was carried out in a Go Taq Flexi PCR (Promega, Madison, WI, USA) reaction mixture containing 2 µL of DNA template, 0.4 µL of primers, 4 µL of 5× PCR buffer, 1.6 µL of 25 mM MgCl_2_, 0.4 µL of dNTPs, 0.2 µL of Taq DNA polymerase, and 11 µL of distilled water (ddH_2_O). Following an initial denaturation at 94 °C for 4 min, 35 cycles of denaturation at 94 °C for 45 s, annealing at 55 °C for 35 s, and extension at 72 °C for 1 min were performed, followed by a 10 min extension at 72 °C. Finally, the purified PCR products were forwarded to a local laboratory company (First BASE Laboratories Sdn Bhd) for Sanger sequencing of the COI gene of the mitochondrial DNA (mtDNA-COI). In addition, 17 COI sequences of terrapin were extracted from GenBank and downloaded, while five COI sequences of terrapin were extracted from the BOLD Systems. This analysis led to the discovery of four novel sequences (GenBank accession numbers: OL658844–OL658847) for 26 sequences (Table 1).

### 2.3. DNA Barcode Sequence Quality Control Measures and Analysis

Chromatograms displaying the nucleotide sequences of both DNA strands for each sample were created—trimmed chromatograms with more than 2% unclear bases and low-quality noisy sequences on both ends. The bidirectional reads were eliminated by benchmarking against a quality value greater than 40. The consensus sequences were obtained by combining the forward and reverse chromatograms in SeqScape, version 2.7 (Applied Biosystems), and comparing them with reference sequences from the NCBI nucleotide (NT) database using BLASTn [38,39]. Additionally, using our COI sequences in a BLASTn search of GenBank, the species that most closely matched our sequences were noted. The sequences’ accession codes and BOLD sequence identifiers were confirmed against GenBank and the BOLD Systems (Table 1). Using the BOLD Systems’ sequence analysis [40], the Kimura 2 Parameter (K2P) model was used to calculate the pairwise sequencing divergences for the distance analyses. MEGAX was used to find the polymorphic sites (PS) or variable sites [41].

### 2.4. Analyses of Molecular Phylogenetics and Divergence Times

The best-fitting evolutionary model for each sequence analysed was determined using the Akaike information criterion (AIC) with sample size correction implemented in jModelTest2 on XSEDE (2.1.6) [42]. The phylogenetic studies used models of sequence evolution selected as best with jModelTest2 for coding and non-coding sequences. maximum likelihood (ML) analyses [43] were performed. As a result, the alignments were carried out in MEGAX using ClustalW [41]. All sequences produced multiple alignments with the same length and beginning point. However, Jalview, Ref. [44], was used to accomplish various sequence alignments, functional site analyses, and web postings of alignments between *B. affinis affinis* and *B. baska* [45]. IQ-tree was used for phylogenetic reconstruction by [46] on XSEDE and [47] via the online CIPRES Science Gateway V.3.3 [48]. The trees were visualised in FigTree v1.4.4 [49].

On the other hand, using the BEAST v2.6.6 tool, the phylogenetic tree topology and divergence dates were computed concurrently [50,51]. BEAUti 2 [52] was used to unlink the substitution models of the data partitions and implement the sequence evolution models selected with jModelTest2 as optimal. The “Clock Model” was set to a rigorous clock with uncorrelated rates, while the “Tree Model” was assigned to a Yule speciation process. The sequences were examined using a relaxed molecular clock model, which permits substitution rates to vary among branches based on an uncorrelated lognormal distribution [50]. We established the species tree before the Yule process. Two simultaneous assessments were conducted utilising Bayesian Markov Chain Monte Carlo (MCMC) simulations with a sampling frequency of 5000 for 100,000,000 generations. The nucleotide substitution model for ML was empirically set to TN93. Bootstrap analysis (1000 pseudoreplicates) provided branch support, and all other parameters were left at their default settings.

After that, the phylogenetic trees were plotted using FigTree v1.4.4. To create the phylogenetic trees, the whole mitochondrial COI sequences of *Batagur affinis* (MTD042-21) and the out-group species *Ophiophagus hannah* (MH153655) were chosen from the GenBank online database [33,53]. Then, using the software DnaSP 6.12.03, we analysed the haplotype of each specimen [54,55,56]. A Median Joining (MJ) network analysis by [57] was performed with NETWORK 10.2.

## 3. Results

### 3.1. Taxonomic Range and Red List Coverage

Table 1 contains the details on the taxa used in this study. The final data collection includes 12 species from the Testudines order, two previously unrepresented in the barcode database. One is not available in the BOLD Systems, and five were not sent to GenBank. We initially deposited our novel COI gene of the mitochondrial DNA (mtDNA-COI) samples (*Batagur affinis* ssp.) in the GenBank database portal.

As a result, the IUCN Red List assigned the 12 species of terrapins sampled for this study to the classifications of least concern (LC) for 33% of the samples, critically endangered (CR) for 25% of the samples, vulnerable (VU) for 8% of the samples, and endangered (EN) and near-threatened (NT) for 17% of the samples (Figure 2).

### 3.2. COI Divergence Assessment

All 26 produced barcodes had sequence lengths of more than 503 bp with no indels or stop codons found. The nucleotide composition was as follows: 16.88% Guanine, 27.21% Cytosine, 27.5% Adenine, and 38.41% Tyrosine. GC Codon position 1 was 52.62% followed by GC Codon position 2 (43.21%) and GC Codon position 3 (36.46%). Almost all species (83.33%, ten species) were represented by dual specimens with a single specimen representing another species and five specimens representing another species (Appendix A).

The genetic divergences of the COI sequences within the order Testudines were studied at various taxonomic levels (Table 2). The genetic divergence rose with the taxonomic rank as expected. The hierarchical taxonomic relationship was directly associated with increased K2P genetic divergence. The conspecific K2P levels ranged from 0% to 2.14% with a mean of 0.68% (SE = 0.04). The mean K2P divergence amongst the congeneric species specimens was 5.49% (SE = 0.15; range 0–9.14%). The average K2P divergence between the specimens from various genera in the same family was 17.10% (SE = 0.03; range: 4.98–22.48%). This range, though they overlap, indicates intraspecific (S) and intragenus (G) distances (Appendix A).

Deep intraspecific K2P divergences were identified in a *Batagur baska* (2.14%) that exceeded the conventional threshold distance of 2% [12,58] (Table 3). A barcode gap analysis revealed that practically all species represented by multiple sequences had a barcode gap (Figure 3). Notably, just one species, *Batagur baska*, had its maximum intraspecific and nearest neighbour distances (0%).

### 3.3. Population Relationships

The nucleotide diversity at 199 nucleotide positions and transitions is approximately 55% saturated (Appendix A). When all codon locations are analysed, transitions and transversions are displayed against the pairwise sequence divergence Tajima-Nei Method (TN84) for the terrapins utilising 503bp of the COI DNA barcode (Figure 4). DAMBE [59] uses these substitution models to perform various molecular phylogenetic analyses. DAMBE also includes functions for determining the optimum substitution models for particular sequences.

The network had 16 haplotypes (Figure 5), which were confirmed with DNAsp 6.12.03 analysis (Table 1). Different haplotypes were found in *Malaclemys terrapin*, *Emys orbicularis*, *Melanochelys trijuga*, *Trachemys scripta elegans*, and *Batagur affinis* ssp. Furthermore, *Batagur baska* and *Batagur affinis affinis* shared a single haplotype (Hap_1), which was shown to be the most variable haplotype. The remaining haplotype only had two specimens and one species.

## 4. Discussion

This study examined 26 terrapin COI sequences from the order Testudines. The species are divided into three families: eight Geoemydidae species (18%), three Emydidae species (6%), and one Pelomedusidae species (2%) (Appendix A and Appendix A). Based on the IUCN Red List of the 12 species of terrapins, 25% were critically endangered (CR) and 8% were endangered (EN). The terrapins studied all inhabit fresh or brackish water [26]. Furthermore, “terrapin” refers to more or less aquatic, hard-shelled turtles [60]. Notably, refs. [3,26] identified 13 terrapin species worldwide but ignored a previously thought-to-be-extinct Seychelles black terrapin species (*Pelusios seychellensis*). However, a genetic analysis of the lectotype revealed that this terrapin is not extinct and is now known as *Pelusios castaneus*. Before the Zoological Museum Hamburg bought a private collection of specimens in 1901 [26,61], the specimens could have been mislabelled or mixed up.

Therefore, the discovery of species-specific COI sequences allows for the identification of terrapin species using DNA barcodes to supplement taxonomy. This can also be used in the field when identifying lost nests or those caught as bycatch in fishing nets. When no other material is available, terrapin eggs or meat are used in the forensic investigations [4].

Additionally, DNA barcoding holds excellent promise for species identification and other conservational genetic applications in terrapins, which are distinct in the evolutionary tree of terrapins for inhabiting the river realm and are well-known for their lengthy migrations. One of the main objectives of the DNA barcoding initiative, species identification, was accomplished using their COI sequences. Even though these ancient taxa have undergone relatively slow molecular evolution [62,63], diagnostic sites at the COI gene were found for all 12 species of terrapins. Ref. [9] found that the distance-based analysis of COI sequences always put members of the same species together, even though the phenetic methods required a total baseline sample for a correct assignment. Using distinct nucleotide combinations, unique COI barcodes were generated for each of the 12 previously defined terrapin species (Appendix A). The diagnoses were reliable with species-specific haplotypes [9] (Table 1; Figure 5).

If a phenetic technique based on a BLAST search was used without a comprehensive baseline sample, such as the one available in GenBank prior to this work, query sequences could be assigned to the wrong species. There were no *Batagur affinis* ssp. COI sequences in GenBank, for example, and a query on a Southern River Terrapin (*B. affinis affinis*) grouped it with a Northern River Terrapin (*B. baska*). The BLASTn search validated it, showing 100% similarity between the *B. affinis*-MTD042-21 COI sequences and *B. baska*-HQ329671 COI sequences (Table 1). So, Jalview’s alignment and visualisation (Figure 6) showed that the sequences of *B. baska* (GenBank Accession Number: HQ329671) and *B. affinis* (BOLD ID: MTD042-21) were very similar. Similarly, *Emys orbicularis*, a species with COI sequences in GenBank, may be confused with *Emys trinacris* or a cryptic species due to 98% identical COI BLASTn results (Table 1).

Furthermore, in the BOLD Systems, the identical sequence of the Northern River Terrapin has two different BIN numbers (AAW2850 and ADX0374), which could be misinterpreted as Southern River Terrapin or a cryptic species.

The detection of the so-called “barcode gap,” which can be measured by comparing the highest intraspecific distance with the minimum interspecific distance (also known as the nearest neighbour genetic distance), is one of the premises of DNA barcoding [64]. Moreover, DNA barcodes are helpful in the investigation of cryptic species [65], particularly those that appear similar but differ genetically [66]. A morphological species gap is strong evidence for species-level cryptic diversity [67]. On the other hand, the absence of a gap between two morphological species implies that they are different forms within the same species, or that they share ancestral polymorphism and/or hybridisation followed by introgression. In this case, it would be helpful to use a multigene (i.e., genomic) method to figure out the reciprocal taxonomic status of the two morphological species [68].

Table 3 shows that the DNA barcoding method revealed possible hidden variety within a species while failing to discover a meaningful difference between two biological species (*B. baska* and *B. affinis*). Such findings demand additional taxonomic research. In comparison to the mean congeneric divergence (5.49%), the mean conspecific K2P divergence (0.68%) was eight times smaller. Thus, as predicted, there was less genetic diversity between the conspecific individuals than between the congeneric species. It makes sense that there would be a rise in the taxonomic levels and an increase in the genetic divergence [69]. Therefore, both mean genetic estimations are comparable to those that have already been noted. In most fish molecular analyses, the conspecific divergence was found to be 0.25–0.39%, while the congeneric divergence was found to be 4.56–9.93% [70,71,72,73,74].

### 4.1. Population Relationships

This research began by examining the terrapins’ DNA barcodes and mitochondrial COI gene haplotypes worldwide. Some existing terrapins and sea turtles are reported to carry mitochondrial COI gene haplotypes [4,9,26,30]. Nonetheless, our study contributes significantly by discovering new sequences from previously unknown areas in Malaysia and around the world. Previous research employing the COI gene in DNA barcoding of terrapins and sea turtles identified 1–10 haplotypes [4,9,30]. This study revealed 16 haplotypes (Table 1; Figure 5) of terrapins from around the world. The BOLD Systems differ from those previously described in Bota Kanan, Perak, and Kuala Berang, Terengganu. Also, the novel *B. affinis* ssp. COI gene sequences from Malaysia were submitted to GenBank (Table 1). They may serve as a reference for future genetic research of populations. A more comprehensive analysis involving additional sites and samples will be necessary to find common haplotypes. Previous studies by [28,75] described the divergence of *Batagur baska* and *Batagur affinis* ssp. Our research checks the sequences between the Indian and Malaysian populations. Moreover, the sequences from the Malaysian specimens are novel, and we hypothesise that this population is exclusive to this region (Figure 1).

Thus, clustering analyses and haplotype networks indicate that the three families are separated by four significant unique lineages (Figure 7). Figure 5 demonstrates that Hap_1 and Hap_16 are more closely related than other haplotypes. Hap_1 contains two *B. baska* specimens and two *B. affinis affinis* specimens, while Hap_16 contains three *B. affinis edwardmolli* specimens, which are in line with [14] that only found a haplotype in the Kuala Berang, Terengganu population; it has been proven that this is a random sampling, and we are not focusing on a clutch. In this case, it appears to be a cryptic species between *B. baska* and *B. affinis affinis*. We would need a more extensive set of genes and many markers from the nuclear genome [66,76,77] to decide if these groups should be called species or subspecies. Perhaps revision is required following the separation of *B. baska* and *B. affinis* ssp. by [28,75]. Even though it can be challenging to identify the morphological diagnostic features in morphologically cryptic species [78,79], the usefulness of such diagnoses may be in doubt [80]. We now recognise that cryptic species are relatively abundant [81,82] and widespread across most animal phyla [83,84]. Moreover, recent DNA research discovered cryptic species in many aquatic taxa [85], raising the possibility that aquatic biodiversity is higher and speciation possibilities have occurred more frequently than previously thought [86].

In addition, using Bayesian analysis, the maximum likelihood phylogeny of the investigated dataset revealed coherent, monophyletic clustering of all studied species (Figure 7). On the phylogenetic tree, cohesion was also detected between the database reference sequences for the representative species and the created sequences. The species were classified according to their family with Geoemydidae being the most abundant. The evolutionary tree indicates that *B. baska* originated in India and is closely related to *B. affinis affinis* from Malaysia, which is supported as a potential cryptic species. *Melanochelys trijuga* is similar to the Persian Gulf’s *Mauremys caspica*, but the *Malaclemys terrapin* in North America is identical to *Trachemys scripta elegans*.

### 4.2. Conservation Status

The International Union maintains the Red List for biodiversity for the Conservation of Nature (IUCN). The IUCN is essential for guiding and igniting conservation and policy change activities; it is much more than a list of species and their states. The preservation of the natural resources that humans depend on is essential [87,88]. The IUCN Red List Categories and Criteria are designed to offer a clear framework for locating species in danger of going extinct globally. According to [87], species can be “Not Evaluated,” “Data Deficient,” “Least Concern,” “Near Threatened,” “Vulnerable,” “Endangered,” “Critically Endangered,” “Extinct in the Wild,” or “Extinct”.

Nearly every nation with native species has its own conservation effort (Table 4). Three *Batagur* species of terrapin, *B. affinis*, *B. baska*, and *B. borneoensis*, are listed as having Critically Endangered (CR) status in Table 1. Moreover, *B. affinis* ssp. falls under the Extinct in the Wild (EW) category in Southeast Asian nations, including Indonesia, Singapore, Thailand, and Vietnam [23,25]. *B. affinis* ssp. is currently restricted to Malaysia and Cambodia. Ref. [89] also states that *B. baska* may be threatened in Thailand and Myanmar. Additionally, *B. borneoensis* was discovered in Brunei, Malaysia, and Indonesia, although it was virtually extinct in Thailand [90].

## 5. Conclusions

In conclusion, COI remains an effective barcode marker for terrapin species, contributing vital information that can be utilised to distinguish and identify genera and species. Compatibility with traditional taxonomy could provide a solid and dependable instrument for accurate species identification and biodiversity assessment facilitation. However, more markers and specimens from new sites should be added to the collection to more accurately compare terrapin populations. The detailed results provided fresh insights into the taxonomic classification of terrapins and revealed the existence of potential cryptic species. This investigation found compelling evidence of potential cryptic species between *B. baska* and *B. affinis affinis*. Our research shows that *B. affinis affinis* might be the same species as *B. baska*, but *B. affinis edwardmolli* might be its own species. However, further research is required. Therefore, the genomic and bioinformatics analysis of terrapins described here could serve as a reference for future global studies of this species and permit a more rational attempt to conserve terrapins. The proposed conservation units are based on the fact that phylogeny and phylogeography change over time and space.

## Figures and Tables

**Figure 1 animals-13-01720-f001:**
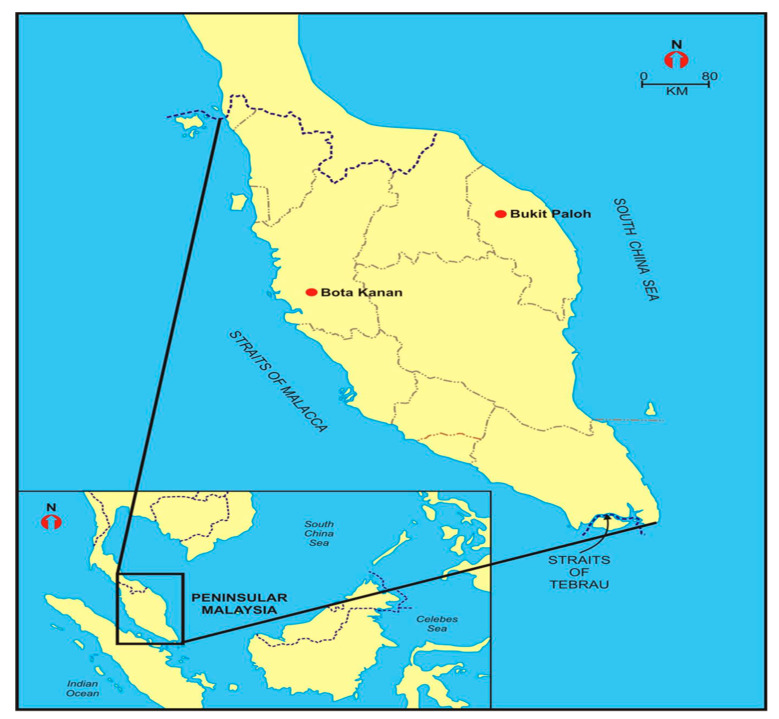
Sampling sites of *Batagur affinis* ssp.

**Figure 2 animals-13-01720-f002:**
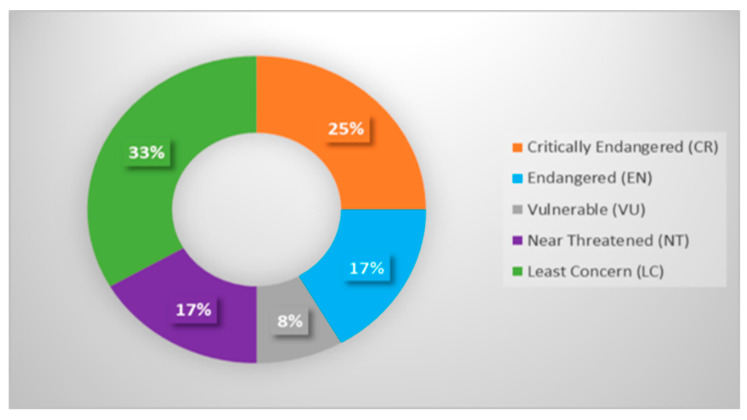
The conservation status of the terrapins is based on the IUCN Red List.

**Figure 3 animals-13-01720-f003:**
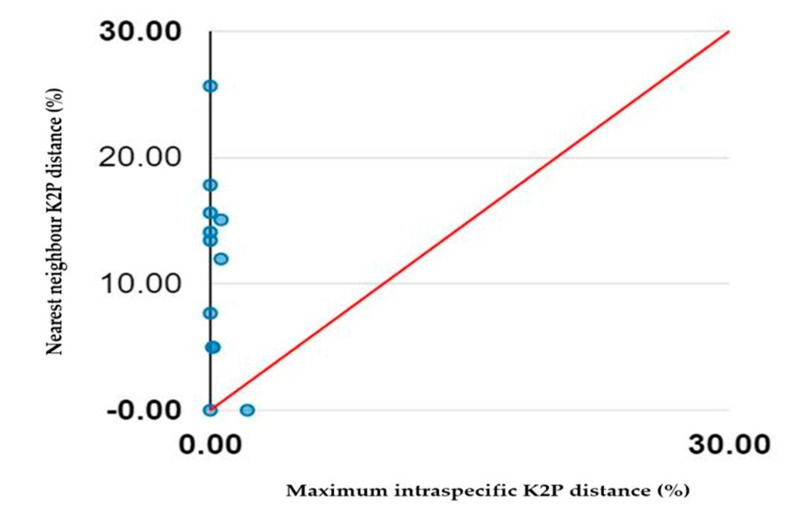
Maximum intraspecific distances plotted against nearest neighbour distances.

**Figure 4 animals-13-01720-f004:**
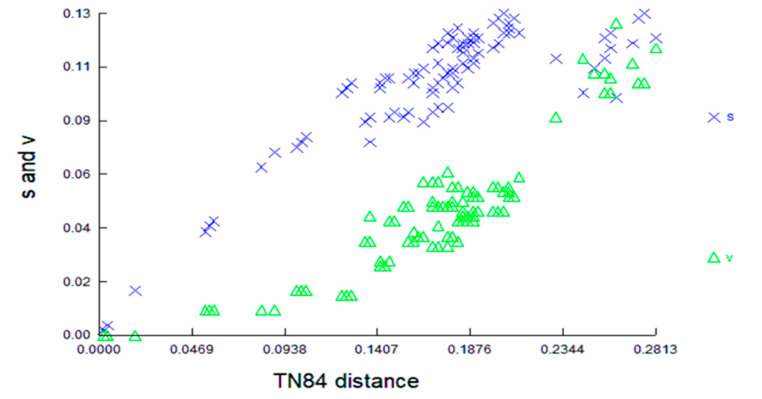
Transitions and transversions are plotted against the pairwise sequence divergence using the Tajima-Nei Method for the terrapins using 503 bp of the COI DNA barcode.

**Figure 5 animals-13-01720-f005:**
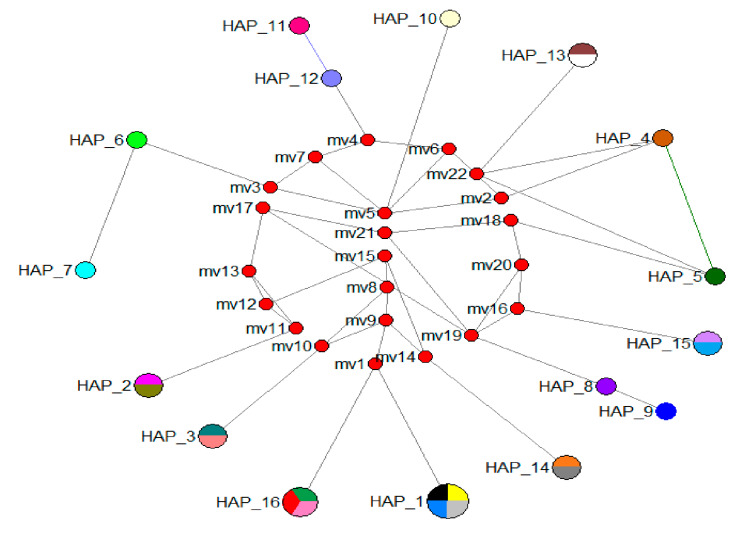
Median-joining network of mtDNA COI haplotypes in the terrapins. The sizes of the circles are proportional to the haplotype frequencies, and the colour-coding corresponds to the locations. The black squares on the lines linking the haplotypes represent the number of mutations.

**Figure 6 animals-13-01720-f006:**
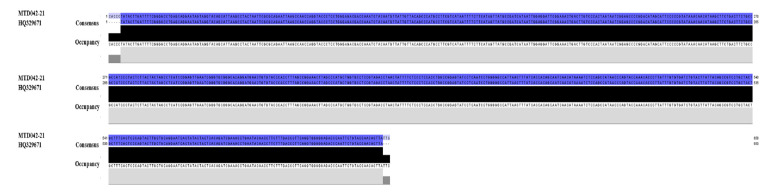
*B. baska* with GenBank Accession Number HQ329671 vs. *B. affinis* BOLD ID MTD042-21 alignment and visualisation with Jalview.

**Figure 7 animals-13-01720-f007:**
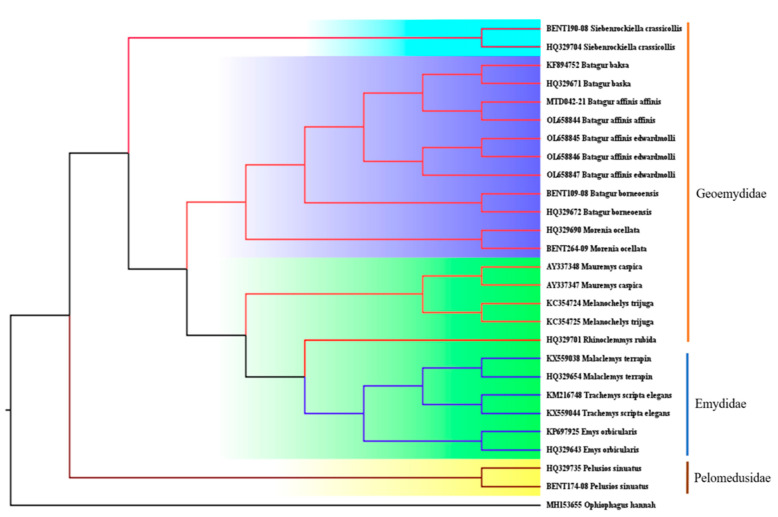
A maximum likelihood tree was constructed with a Bayesian analysis based on the COI sequences belonging to the order Testudines.

**Table 1 animals-13-01720-t001:** List of terrapin species studied through DNA barcoding with the BOLD IDs of their respective COI sequences and the GenBank accession of each species.

Scientific Name	English Name	GenBank	BOLD ID	Haplotype	BLASTn Result	Locallity	IUCN Red List	References
*Batagur baska*	Northern River Terrapin	KF894752	GBGCR2852-19	Hap_1	100% with *B. affinis* (OL658844)	India	CR	[31]
*Batagur baska*	Northern River Terrapin	HQ329671	GBGCR2716-19	Hap_1	99% with *B. affinis* (OL658844)	India	CR	[32]
*Batagur borneoensis*	Painted Terrapin	HQ329672	GBGCR2717-19	Hap_2	95% with *B. trivittata* (HQ329675)	Indonesia	CR	[32]
*Batagur borneoensis*	Painted Terrapin	None	BENT109-08	Hap_2	95% with *B. trivittata* (HQ329675)	Indonesia	CR	[33]
*Morenia ocellata*	Bengal Eyed Terrapin	HQ329690	GBGCR2724-19	Hap_3	90–91% with *M. petersi* (MH157788)	Myanmar	EN	[32]
*Morenia ocellata*	Bengal Eyed Terrapin	None	BENT264-09	Hap_3	90–91% with *M. petersi* (KF894774)	Myanmar	EN	[33]
*Malaclemys terrapin*	Diamondback Terrapin	HQ329654	GBGC11262-13	Hap_4	95% with *Graptemys barbouri* (MG728234)	America	VU	[32]
*Malaclemys terrapin*	Diamondback Terrapin	KX559038	GBGCR2938-19	Hap_5	95% with *Graptemys geographica* (MG728245)	America	VU	[34]
*Emys orbicularis*	European Pond Terrapin	HQ329643	GBGC11273-13	Hap_6	98% with *E. trinacris* (KX559027)	Unknown	NT	[32]
*Emys orbicularis*	European Pond Terrapin	KP697925	None	Hap_7	98% with *E. trinacris* (KX559027)	Germany	NT	[35]
*Melanochelys trijuga*	Indian Pond Terrapin	KC354725	GBGC11418-13	Hap_8	96% with *M. tricarinata* (KF894770)	India	LC	[31]
*Melanochelys trijuga*	Indian Pond Terrapin	KC354724	GBGC11419-13	Hap_9	95% with *M. tricarinata* (KF894770)	India	LC	[31]
*Rhinoclemmys rubida*	Mexican Spotted Terrapin	HQ329701	GBGCR2766-19	Hap_10	91% with *R. annulata* (MH274599)	Mexico	NT	[32]
*Trachemys scripta elegans*	Red-eared Terrapin	KX559044	GBGCR1038-18	Hap_11	96–100% with *T. s. elegans* (TSU49047)	America	LC	[34]
*Trachemys scripta elegans*	Red-eared Terrapin	KM216748	GBGCR1008-15	Hap_12	97–100% with *T. s. elegans* (TSU49047)	America	LC	[36]
*Pelusios sinuatus*	Serrated Hinged Terrapin	None	BENT174-08	Hap_13	100%	Southern Africa	LC	[33]
*Pelusios sinuatus*	Serrated Hinged Terrapin	HQ329735	GBGC11221-13	Hap_13	100%	Southern Africa	LC	[32]
*Siebenrockiella crassicollis*	Smiling Terrapin	HQ329704	GBGCR2769-19	Hap_14	100%	Unknown	EN	[32]
*Siebenrockiella crassicollis*	Smiling Terrapin	None	BENT190-08	Hap_14	100%	Unknown	EN	[33]
*Mauremys caspica*	Striped-neck Terrapin	AY337348	GBGC0806-06	Hap_15	95% with *Chinemys nigricans* (AF348264)	Iran	LC	[37]
*Mauremys caspica*	Striped-neck Terrapin	AY337347	GBGC0805-06	Hap_15	95% with *Chinemys nigricans* (AF348264)	Bahrain	LC	[37]
*Batagur affinis*	Southern River Terrapin	None	MTD042-21	Hap_1	100% with *B. baska* (HQ329671)	Malaysia	CR	[33]
*Batagur affinis affinis*	Southern River Terrapin	OL658844	HYT001-21	Hap_1	99–100% with *B. baska* (KF894752)	Malaysia	CR	This study
*Batagur affinis edwardmolli*	Southern River Terrapin	OL658845	HYT002-21	Hap_16	98% with *B. baska* (KF894752)	Malaysia	CR	This study
*Batagur affinis edwardmolli*	Southern River Terrapin	OL658846	HYT003-21	Hap_16	98% with *B. baska* (KF894752)	Malaysia	CR	This study
*Batagur affinis edwardmolli*	Southern River Terrapin	OL658847	HYT004-21	Hap_16	98% with *B. baska* (KF894752)	Malaysia	CR	This study

**Table 2 animals-13-01720-t002:** K2P divergence values from the examined specimens of varying taxonomic levels. SE = standard error.

Category	n	Taxa	Comparisons	Min (%)	Mean (%)	Max (%)	SE (%)
Within Species	25	11	20	0	0.68	2.14	0.04
Within Genus	9	1	24	0	5.49	9.14	0.15
Within Family	24	2	125	4.98	17.10	22.48	0.03

**Table 3 animals-13-01720-t003:** The summary statistics include the BIN of each species, their maximum intraspecific K2P distances, and the nearest neighbour K2P distances (i.e., minimum interspecific distance).

Scientific Name	BIN	Nearest Species	Max. Intraspecifc Distance (%)	Nearest Neighbour Distance (%)
*Emys orbicularis*	BOLD:AAF8183	*Malaclemys terrapin*	0.62	11.97
*Malaclemys terrapin*	BOLD:AAX3718	*Trachemys scripta*	0.16	4.98
*Trachemys scripta*	BOLD:AAF5910	*Malaclemys terrapin*	0.14	4.98
*Batagur affinis*	BOLD:AAW2850 & ADX0374	*Batagur baska*	2.14	0
*Batagur baska*	BOLD:AAW2850	*Batagur affinis*	0	0
*Batagur borneoensis*	BOLD:AAW2847	*Batagur affinis*	0	7.67
*Mauremys caspica*	BOLD:AAJ1604	*Malaclemys terrapin*	0	14.09
*Melanochelys trijuga*	BOLD:AAX4497	*Mauremys caspica*	0.62	15.07
*Morenia ocellata*	BOLD:AAX4362	*Batagur borneoensis*	0	13.43
*Rhinoclemmys rubida*	BOLD:AAY0332	*Malaclemys terrapin*	0	15.62
*Siebenrockiella crassicollis*	BOLD:AAJ6683	*Batagur borneoensis*	0	17.82
*Pelusios sinuatus*	BOLD:AAX1329	*Batagur affinis*	0	25.66

**Table 4 animals-13-01720-t004:** Conservation centre records for *Batagur* sp. in indigenous species country.

Scientific Name	Common Name	Conservation Centre	Country
*Batagur affinis*	Southern River Terrapin	Sre Ambel River, Koh Kong Reptile Conservation Centre	Cambodia
*Batagur affinis*	Southern River Terrapin	Angkor Center for Conservation of Biodiversity	Cambodia
*Batagur affinis*	Southern River Terrapin	Kedah River, Kepala Batas, Kedah	Malaysia
*Batagur affinis*	Southern River Terrapin	Perak River, Bota Kanan, Perak	Malaysia
*Batagur affinis*	Southern River Terrapin	Terengganu River, Kuala Berang, Terengganu	Malaysia
*Batagur affinis*	Southern River Terrapin	Kemaman River, Kemaman, Terengganu	Malaysia
*Batagur affinis*	Southern River Terrapin	Setiu Wetlands State Park, Terengganu	Malaysia
*Batagur baska*	Northern River Terrapin	Vawal National Park	Bangladesh
*Batagur baska*	Northern River Terrapin	Sajnekhali Forest Station	India
*Batagur borneoensis*	Painted Terrapin	Langkat, North Sumatera	Indonesia
*Batagur borneoensis*	Painted Terrapin	Ujung Tamiang, Aceh	Indonesia
*Batagur borneoensis*	Painted Terrapin	Setiu Wetlands State Park, Terengganu	Malaysia
*Batagur borneoensis*	Painted Terrapin	Pengkalan Balak, Melaka	Malaysia

## Data Availability

The data presented is authentic and was not improperly selected, manipulated, enhanced, or fabricated. The data can be found in GeneBank (accession numbers OL658844-OL658847) and BOLD Systems (Sequence IDs HYT001-21 to HYT004-21).

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
