# Peer review of "Global Terrapin Character-Based DNA Barcodes: Assessment of the Mitochondrial COI Gene and Conservation Status Revealed a Putative Cryptic Species"

_animals, 2023, doi:10.3390/ani13111720_

Round 1

Reviewer 1 Report (Previous Reviewer 1)

The manuscript entitled " Global Terrapin Character-based DNA Barcodes: Assessment of the Mitochondrial COI Gene and Conservation Status Revealed a Putative Cryptic Species" with the manuscript ID of animals-219522 evaluated 26 sequences of terrapins worldwide including 12 species of Malaysia's southern river terrapin populations through COI DNA barcoding and phylogenetic analysis, this work provided fresh insights into the taxonomic classification of terrapins. This manuscript has greatly improved after revision, however, there was still minor issue, which need addressed before the manuscript can be accepted.

Minor issue:

the style of the references should keep consistent from fist to last. for example, in some reference, there was semicolon among the authors, but some do not. Some journal names are abbreviations, but some others are full names, please check them.

Author Response

Dear Prof/Dr.,

REVIEWER’S COMMENTS

Thank you very much for reviewing my manuscript. I also greatly appreciate the reviewers' insights and suggestions to improve this paper. I have carried out the reviewer's suggestions and revised the manuscript accordingly.

2. I've attached the manuscript; an amendment has been made. Please kindly refer to line 525 (references chapter). I hope that you find our responses satisfactory.

Thank you.

An amendment has been made. Please kindly refer to line 525 (references chapter). Thank you. 

Reviewer 2 Report (Previous Reviewer 2)

The authors have provided reasonable responses to my concerns (see Animals Reviewer 2 Author Responses, attached). I suggest that brief summaries of these statements be included in the Discussion to acknowledge potential limitations of the study, and to provide information that may be helpful for study replication or follow-up research.

Author Response

Dear Prof/Dr.,

REVIEWER’S COMMENTS

Thank you very much for reviewing my manuscript. I also greatly appreciate the reviewers' insights and suggestions to improve this paper. I have carried out the reviewer's suggestions and revised the manuscript accordingly.

2. I've attached the manuscript. The amendment has been highlighted in yellow. I hope that you find our responses satisfactory.

Thank you.

This manuscript is a resubmission of an earlier submission. The following is a list of the peer review reports and author responses from that submission.

Round 1

Reviewer 1 Report

The manuscript entitled " A Novel COI Gene of Batagur affinis ssp. inferred by DNA Barcode Analysis Reveals a Potential Cryptic Species" with the manuscript ID of animals-1900917 evaluated the use of COI DNA barcodes in 12 species of Malaysia's Southern river terrapin populations to understand better the species' genetic divergence and other genetic characteristics. These species were divided into three families by using DNA barcodes, which are an excellent way to measure the diversity of a population. This work is meaningful, and provid fresh insights into the taxonomic classification of terrapins. However, there are a few issues that should be addressed before it can be accepted.

Major issue:

The language of this paper should be checked and reedited.

Minor issues:

1)   Line 50, please provide the Latin name for the “bird species”. You can give some examples.

2) Line 158, Change “are” into “were”.

3) Line 251-254, This sentence is too vaguely, please rewrite it.

4) Line 424: Please check the style of the references, especially journal name

Reviewer 2 Report

The authors used DNA barcoding to evaluate 26 COI sequences among 12 terrapin species. The manuscript is generally well-written, and the literature review is thorough. Some of the results could have important conservation implications for several critically endangered Batagur species.

My primary concern with the study is the very small number of samples analyzed. Only one blood sample was obtained from B. affinis affinis, and only three samples were obtained from B. affinis edwardmolli. Further, the B. affinis affinis sample was provided by a captive breeding facility, so there may be some uncertainty regarding the genetic origin of that sample. It’s also unclear if the three B. affinis edwardmolli samples are from hatchlings from the same clutch. It seems reasonable to expect that a barcoding and phylogenetic analysis of species would include samples from at least three individuals of each species that are not siblings.

Reviewer 3 Report

Overall, I think this is interesting work, but the logic behind parts of it are hard to follow.

Firstly, the paper is titled about a novel COI gene for B. affinis and how there could be cryptic speciation, but this does not seem to be the crux of the work. I think it would be more suitable to title it something like “Characterization of the COI gene for terrapins globally,” as that is what the paper is mostly geared towards.

Secondly, and what I see as a really confusing aspect of this manuscript, is that this study is centered around terrapins, which are a polyphyletic group of turtles. This is addressed in line 44, but it is not articulated as to why the study focuses exclusively on terrapins. Additionally, in Table 1, a number of the species listed by their English name are not in fact terrapins. For instance the Trachemys scripta, Morenia ocellata, Emy orbicularis, Melanochelys, and others Are freshwater turtles. Therefore, it is my recommendation that this manuscript lose the pretenses that this is a study on terrapins, and this is in fact a study on turtles and terrapins across multiple families and suborders.

Though the English is very good, there are still some errors and areas of confusion. Line 58 “ complicated accessible anatomy” is confusing. Line 60: have these genes been used in eDNA, or how were they used to find turtles? Line 63: define geographically “the north”. Line 83, remove so, replce “find out” with and replace with “determine”. Line 86, remove “also”.

Line 103 – put permit # in acknowledgements

Line 121: Madison, WI, USA

Remove Figure 2. It is not necessary, and in the text, give the number of species for each conservation classification, not the percentage. This is hard to read and follow without knowing how many species were actually tested.

Line 202 “another species” not “a species”

Line 255, remove edible, this is not necessary information. given the imperiled nature of these turtles, we should not advocate for their consumption.

A timeline needs to be added to figure 7 to indicate just how deeply divergent the putative cryptic species of Batagur affinis is.

Overall, I think the paper is weak and many regards. Most notably, I think it is weakest and comparing distantly related groups of turtles under the guise of “terrapins,” especially when many turtles in the analysis are not terrapins.